# A Reinforced, Event-Driven, and Attention-Based Convolution Spiking Neural Network for Multivariate Time Series Prediction

**DOI:** 10.3390/biomimetics10040240

**Published:** 2025-04-13

**Authors:** Ying Li, Xikang Guan, Wenwei Yue, Yongsheng Huang, Bin Zhang, Peibo Duan

**Affiliations:** 1School of Software, Northeastern University Shenyang, Shenyang 110167, China; liying@mail.neu.edu.cn (Y.L.); 20202456@stu.neu.edu.cn (X.G.); 2371447@stu.neu.edu.cn (Y.H.); zhangbin@mail.neu.edu.cn (B.Z.); 2State Key Lab of Integrated Services Networks, Xidian University, Xi’an 710071, China; wwyue@xidian.edu.cn

**Keywords:** spiking neural network (SNN), attention mechanism, multivariate time series prediction, pooling strategy, coding, deep learning

## Abstract

Despite spiking neural networks (SNNs) inherently exceling at processing time series due to their rich spatio-temporal information and efficient event-driven computing, the challenge of extracting complex correlations between variables in multivariate time series (MTS) remains to be addressed. This paper proposes a reinforced, event-driven, and attention-based convolution SNN model (REAT-CSNN) with three novel features. First, a joint Gramian Angular Field and Rate (GAFR) coding scheme is proposed to convert MTS into spike images, preserving the inherent features in MTS, such as the temporal patterns and spatio-temporal correlations between time series. Second, an advanced LIF-pooling strategy is developed, which is then theoretically and empirically proved to be effective in preserving more features from the regions of interest in spike images than average-pooling strategies. Third, a convolutional block attention mechanism (CBAM) is redesigned to support spike-based input, enhancing event-driven characteristics in weighting operations while maintaining outstanding capability to capture the information encoded in spike images. Experiments on multiple MTS data sets, such as stocks and PM2.5 data sets, demonstrate that our model rivals, and even surpasses, some CNN- and RNN-based techniques, with up to 3% better performance, while consuming significantly less energy.

## 1. Introduction

Spiking neural networks (SNNs) are directly inspired by the information encoding mechanisms of biological neural systems, which rely on discrete spiking signals to transmit dynamic spatio-temporal information [1,2,3]. The design of their spiking neurons mimics the biological process of membrane potential accumulation and spiking emission, thereby achieving an event-driven characteristic similar to that of biological brains. This biomimetic property enhances the robustness of the model in complex environments, provides theoretical support for developing novel neuromorphic computing architectures, and it has led to SNNs being recognized as the third generation of neural networks [4,5].

Research on SNNs has revealed that they have a performance comparable to that of conventional artificial neural networks (ANN) while being more energy-efficient [6,7,8,9], particularly in image analysis tasks [10,11,12,13]. However, there has been relatively little prior work on SNNs in relation to MTS analysis, which is essential in our daily lives but limited by energy expenditure, such as portable medical devices for healthcare that monitor various physical signs [14,15,16,17].

Recently, SNN-based solutions to univariate time series (UTS) analysis have been of interest to researchers [18]. To this end, advanced techniques from conventional ANN models, such as convolution neural networks (CNN) and recurrent neural networks (RNN), have been incorporated into the architecture design of SNN models [19,20,21,22,23]. For example, spike-based memory cells have been proposed by mimicking gate mechanisms in long-short-term memory (LSTM) and gated recurring unit (GRU) setups, which have the ability to analyze temporal patterns in UTS [21,24,25,26]. Convolution layers have also been introduced in SNNs [27], taking advantage of the ability of convolutional kernels to extract temporal patterns in a time series matrix.

Unfortunately, it is still difficult to use the SNN-based approaches mentioned earlier to analyze multivariate time series (MTS). This is mainly attributed to the complex spatio-temporal correlations posed by the high-dimensional nature of MTS [28,29]. Motivated by the convolution SNN (CSNN) architecture proposed by Gautam et al. [27], our aim is to develop a variant of CSNN to address the MTS prediction issue by overcoming the following challenges.

First, existing encoding schemes have a limited ability to accurately represent knowledge that corresponds to different classes of characteristics in MTS [30]. It further results in the performance bottleneck of existing SNN-based methods [31]. To solve this problem, Long et al. [32] employed the non-subsampled shearlet transform (NSST) technique. However, the state of a neuron was denoted as a real value, rather than as a spike.

Second, neither max-pooling nor average-pooling is effective enough to reduce the dimension while still preserving features of spike images, as they differ from regular images [33,34,35]. More precisely, max-pooling can lead to an overload of spike activation, further resulting in more energy consumption, while average polling has difficulty in threshold configuration, which can cause information loss.

Third, the spike-based memory cell has the weakness of hardly achieving the energy-efficient sparse firing regime [36] or losing the nugget of information [37]. The transformation of MTS into spike images in convolution-based SNN increases the dimension of the data, resulting in the discontinuity and randomness of the feature distribution in high-dimensional, spike-based images. Although attention mechanisms are potential solutions, most of them are implemented based on real values rather than spikes, which violates the integrated event-driven nature of SNNs.

In an effort to deal with the above problems, this paper presents a reinforced, event-driven, and attention-based convolution SNN model (REAT-CSNN), which integrates an effective spike-encoding scheme and a novel pooling strategy. A variant of the attention mechanism is developed to deal with the critical spaito-temporal correlations from spike images, which have high dimensions due to the encoding. More precisely, the main contributions are as follows.

By incorporating the advantages of Gramian Angular Field (GAF) and rate encoding techniques, the proposed spike-encoding strategy allows for an effective representation of the knowledge with respect to temporal patterns and variable dependencies in MTS.The proposed Leaky-Integrate-and-Fire (LIF)-based pooling strategy has been theoretically and experimentally proven to be effective in extracting more features from the regions of interest in a spike image than average-pooling strategies.The redesigned Spike-based Convolutional Block Attention Mechanism (SCBAM) strengthens its event-driven characteristics in weighting operations while still having the outstanding capability to capture the spatio-temporal correlations encoded in the spike images.By performing experiments on multiple MTS data sets, the results show that the performance of our model rivals, and even surpasses, some CNN-, RNN-, and SNN-based techniques, with up to 3% better performance but with less energy consumption.

## 2. Related Work

As of late, the progress of deep learning techniques has prompted the development of models for MTS prediction. These techniques improve prediction performance by capturing the multifaceted nature of nonlinearity, nonstationarity, and spatio-temporal correlation between MTS. In the following, we diagram a portion of the latest and typical methods in MTS prediction methods from the point of view of ANN-based models and SNN-based models.

### 2.1. ANN-Based Models

ANN-based models can be divided into two categories: Euclidean space-based models, such as CNN and RNN; and graph-structure-based models, such as graph neural network (GNN) [38,39,40]. CNNs are commonly used for the analysis of two-dimensional (2D) images. To apply CNNs to time series analysis, existing models attempt to represent MTS as a multidimensional tensor from temporal and spatial dimensions so that convolutional kernels can be used to extract temporal patterns and dependencies. For example, Chen et al. [38] modeled MTS in a traffic network as a 3D volume and then used spatio-temporal kernels in different layers to capture temporal and spatial correlations. Wang et al. [39] modeled MTS as a 2D volume but employed multiple CNNs to address the issue of periodic MTS prediction, and they automatically extracted correlations between different variables from a global perspective. Moreover, 2D volume was also used in [40] with a proposed attention module to learn the representation of spatio-temporal relationships.

Compared to CNN-based models, RNN and the corresponding variants, such as LSTM and GRU [41,42,43,44], are used more frequently in MTS analysis because of their ability to exhibit temporal dynamic behavior in a textual sequence, which has the same 1D structure as time series. One challenge in RNN-based methods is how to use historical information effectively in forecasting future time series. To address this, attention mechanisms are often used. For example, both Qin et al. [41] and Liu et al. [42] developed a dual-stage, attention-based RNN to capture long-term temporal dependencies and to select relevant MTS for predictions. The difference was due to the fact that the former study aimed at single-step prediction, whereas the latter study focused on multiple-step prediction. Similarly, a dual-stage attention mechanism was also employed in the GRU for the MTS prediction developed by Tan et al. [45]. Additionally, the combination of CNN and RNN models has become a popular approach for MTS prediction, such as the Conv-LSTM network model proposed by Xiao et al. [43] and the temporal self-attention-based Conv-LSTM proposed by Fu et al. [44], which benefit from integrating dependencies between time series in CNN and exploring long-term temporal correlation in RNN.

Rather than ANN-based models, GNNs allow for the input of graph structures, which can be used to represent the implicit and explicit correlations between MTS, such as the relationships between the stakeholders of a certain stock. This provides GNNs with the capability to explore the dependencies between time series in depth. For example, Chen et al. [46] proposed a dual CNN and GNN model for stock prediction, which captures spatial and temporal correlations across time series. To further enhance the GNN’s capacity to remember significant features in time series, Rao et al. [37] combined GNN with LSTM. The GNN was used to balance the contribution of the external features of other time series on the target time series, while two parallel LSTMs were used to learn the temporal patterns and MTS dependencies in the long-term history.

### 2.2. SNN-Based Models

As SNN techniques matured, increasing attention has been drawn to solving time series problems using SNN-based approaches. To improve the practicability of these methods, much effort has been dedicated to the development of the spike encoding scheme and the extraction of time series patterns.

The spike encoding scheme allows time series to be represented as spike trains while preserving important information that could be beneficial for accurate analysis. Sharma et al. [20] proposed a continuous firing temporal coding scheme to represent the temporal correlations in a high-dimensional chaotic time series of electricity prices. However, this encoding scheme is only applicable to univariate time series, which restricts its capacity to capture the spatio-temporal correlation in MTS. To address this issue, Long et al. [32] proposed a NSST technique, which enables MTS to be processed in the NSST domain. This makes the SNN model particularly advantageous for capturing the nonlinearity and non-stationarity in MTS, as well as the dependence between time series. Unfortunately, NSST denotes the state of a neuron as a real number, rather than as a spike. Thus, it violates the event-driven nature of SNN.

Regarding pattern extraction, existing studies can be divided into two categories based on the kernel architecture used in SNN models, which are RNN-based or CNN-based. For RNN-based models, Liu et al. [21] developed an SNN by replicating the recurrent structure of RNNs. More precisely, they proposed two gate mechanisms, namely the reset gate and the consumption gate. Similarly, Wachirawit et al. [24] also developed a gated SNN. They managed to solve the issue of non-differentiability for the surrogate derivative and decreased the gap between the derivative of the activation function of the spiking neurons. Yan et al. [47] designed a quantum-inspired SNN that combined a quantum particle swarm optimization algorithm and a Kalman filtering technique to smooth and denoise the original time series data.

Examining the remarkable achievements of SNNs in image analysis, an alternative approach transformed the multivariate time series into matrices, referred to as spike images. This allows models, such as CSNN proposed by Gautam et al. [27], to be used in time series analysis. Powered by convolutional kernels, the features extracted from the spike images correspond to the temporal patterns and dependencies between time series. However, spike images are not the same as practical images, making it difficult to accurately retain useful features without losing information. We propose a novel pooling strategy, in conjunction with a spike-based attention module, to further enhance the feature exploration capacity of CSNN, thereby improving the prediction accuracy.

The success of SNN techniques promotes its practical application in time series analysis. For instance, David et al. [14] presented a novel SNN model for forecasting financial time series, which was capable of recognizing nonstationarity in financial data. Macikag et al. [48] developed a clustering-based ensemble model to predict air pollution, which combined multiple evolving SNNs to analyze a different set of time series related to different air pollution indicators. Capizzi et al. [49] applied an SNN-based method to efficiently and inexpensively capture the temporal dynamics of the chemical processes that occur in the digestor.

Despite the encouraging outcomes of the SNN-based approaches mentioned above, they focus mainly on univariate time series. There are still difficulties to be tackled when dealing with MTS issues, such as spikes encoding, event-driven architecture design, and spatio-temporal extraction, which are the main causes of the considerable gap between SNN and traditional ANN in MTS analysis.

## 3. Preliminary Work

### 3.1. GAF

Gramian Angular Field (GAF) [50] is a technique that can encode a time series in an image while preserving the temporal correlations between time series. Mathematically, we define a time series with observations in the historical *T* steps by $X_{1:T}=\left\{x_{1},x_{2},\ldots ,x_{T}\right\}$. First, we rescale $X_{1:T}$ as ${\tilde{X}}_{1:T}$ with the normalized method in (Equation 1) so that $\forall x_{t}\in X_{1:T}$ falls in the range of [0,1]:(1)$${\tilde{x}}_{t}=\frac{x_{t}-\min X_{1:T}}{\max X_{1:T}-\min X_{1:T}}.$$Next, the ${\tilde{X}}_{1:T}$ in the polar coordinate system is expressed as in the equation in (Equation 2), where the radius is calculated with time step t∈T, and the angular cosine is calculated with $\forall {\tilde{x}}_{t}\in {\tilde{X}}_{1:T}$:(2)$$\left\{\begin{array}{c} r_{t}=\frac{t}{T},t\in T,\\ \theta_{t}=\arccos\left({\tilde{x}}_{t}\right). \end{array}\right.$$

GAF can generate two types of images using the equations of the Gramian angular summation field (GASF) and the Gramian angular difference field (GADF). The major difference between GASF and GADF depends on the conversion of trigonometric functions, where GASF is based on cosine functions and GADF is based on sine functions. In this paper, we employed and defined GASF as follows:(3)$$\left[\begin{array}{cccc} \cos\left(\theta_{1}+\theta_{1}\right) & \cos\left(\theta_{1}+\theta_{2}\right) & \cdots & \cos\left(\theta_{1}+\theta_{T}\right)\\ \cos\left(\theta_{2}+\theta_{1}\right) & \cos\left(\theta_{2}+\theta_{2}\right) & \cdots & \cos\left(\theta_{2}+\theta_{T}\right)\\ \vdots & \vdots & \ddots & \vdots \\ \cos\left(\theta_{T}+\theta_{1}\right) & \cos\left(\theta_{T}+\theta_{2}\right) & \cdots & \cos\left(\theta_{T}+\theta_{T}\right) \end{array}\right].$$

### 3.2. Rate Coding

Rate coding schemes are broadly classified into three categories, including count coding, density coding, and population coding. The proposed SNN model depends on the count rate coding, which has been widely used in SNN [30] to solve image analysis problems.

Typically, the first phase of count rate coding is to duplicate an image $I\in R^{3}$*K* times. Thus, we have $I_{1}=I_{2}=\ldots =I_{K}$. In the second phase, *K* spike images are generated, denoted by ${\tilde{I}}_{1},{\tilde{I}}_{2},\ldots ,{\tilde{I}}_{K}$, where ${\tilde{I}}_{k\in K}$ has the same size as *I*. Each pixel in ${\tilde{I}}_{k}$ is a spike with the value 0 or 1, and they are estimated based on a specific probability distribution that is parameterized with the intensity of the pixel located in the same position in $I_{k}$. For example, Saunders [10] utilized the Poisson distribution with a mean event rate equal to the firing rate that is scaled by the intensity of the pixel.

### 3.3. LIF

The Leaky Integrate-and-Fire (LIF) model is a popularly used SNN model to analyze the behavior of a nervous system. Unlike the conventional artificial neural network, a LIF model consists of multiple layers of LIF neurons. Suppose that the *j*-th LIF neuron in the *l*-th layer receives the spike from the *i*-th neuron in the previous layer at the time step *t*, which is denoted by $s_{i}\left(t\right)$. Furthermore, $s_{i}\left(t\right)$ is modulated with weight $w_{i}$ and then integrated into $c_{j}\left(t\right)$, which behaves like a leaky integrator and decays exponentially according to a constant $\tau_{s}$. Mathematically, we can estimate $c_{j}\left(t\right)$ by the following:(4)$$c_{j}\left(t\right)=\sum_{i=1}^{n^{l-1}}w_{i}s_{i}\left(t\right)+\alpha c_{j}\left(t-1\right),$$
where $n^{l-1}$ is the number of neurons in the layer n−1, α is a constant coefficient, and c(t−1) is the current at the time step t−1.

The dynamic of $c_{j}\left(t\right)$ leads to the variation of the membrane potential $u_{j}\left(t\right)$, which can be described as follows:(5)$$\tau_{m}\frac{du_{j}\left(t\right)}{dt}=u_{rest}-u_{j}\left(t\right)+c_{j}\left(t\right),$$
where $\tau_{m}$ is the time constant of the membrane potential, and $u_{rest}$ represents the resting potential. As shown in Formula (Equation 6), u(t) is reset to $u_{rest}$ once it exceeds the membrane potential threshold $u_{th}$ with the corresponding output $o_{j}=1$. Otherwise, $o_{j}=0$.(6)$$o_{j}\left(t\right)=\left\{\begin{array}{cccc} 1 & , & u_{j}\left(t\right)\geq u_{th}, & u_{j}\left(t\right)=u_{rest}\\ 0 & , & otherwise. \end{array}\right.$$Since it is extremely expensive to estimate $u_{j}\left(t\right)$ through (Equation 5), an approximation of $u_{j}\left(t\right)$ is received using the following iterative expression:(7)$$u_{j}\left(t\right)=u_{j}\left(t-1\right)+\frac{1}{\tau_{m}}\left(u_{rest}-u_{j}\left(t-1\right)+c_{j}\left(t\right)\right).$$

### 3.4. CBAM

Given an intermediate feature map I, the Convolutional Block Attention Module (CBAM) [51] obtains the local interest region of I, which is denoted by $\tilde{I}$, through a channel attention module (CAM) and a spatial attention module (SAM). More precisely, the architecture of CAM is modeled as(8)CAM(I)≜σ(MLP(AvgP(I))+MLP(MaxP(I))),
where MLP is a multiple-layer perceptron, AvgP denotes average-pooling, MaxP denotes the max-pooling, and σ denotes the sigmoid function. Subsequently, SAM is operated on the output of the CAM, $I^{\prime }$, which is expressed as(9)$$SAM\left(I^{\prime }\right)≜\sigma \left(Conv\left(AvgP\left(I^{\prime }\right);MaxP\left(I^{\prime }\right)\right)\right),$$
where Conv represents the convolution operation. $\tilde{I}$ is obtained as follows:(10)$$\begin{array}{c} I^{\prime }=CAM\left(I\right)⊗I, \end{array}$$(11)$$\begin{array}{c} \tilde{I}=SAM\left(I^{\prime }\right)⊗I^{\prime }, \end{array}$$
where ⊗ is the element-wise multiplication.

## 4. Methodology

We modeled a MTS prediction problem as $X_{T+1}=f\left(X_{1:T}\right)$. More precisely, $X_{1:T}=\left\{X_{1},X_{2},\ldots ,X_{T}\right\}$ denotes the MTS in historical *T* time steps. $X_{t\in T}\in R^{N}$ is expressed by $X_{t}=\left\{X_{1,t},X_{2,t},\ldots ,X_{N,t}\right\}$, where *N* denotes the multivariate dimensions. Our objective was to develop a SNN model f(·), named REAT-CSNN, which can predict MTS in the next time step T+1 with $X_{1:T}$. For better clarity, the frequently used abbreviations in this paper are listed in the Abbreviations section.

The architecture of REAT-CSNN is shown in Figure 1. $X_{1:T}$ is first converted to the multichannel spike image series $I=\{I_{1},I_{2},\ldots ,I_{K}\}$ with the proposed Gramian Angular Field Rate coding (GAFR coding) algorithm, which is illustrated in Section 4.1. $I_{k\in K}\in R^{T\times T\times N}$ is a three-dimensional image, in which both the width and height are equal to *T*, and the channel size is *N*.

$I_{k}$ is then transmitted to the LIF-based feature extraction module (LIF-FE) that includes LIF-based convolution (LIF-Conv) and LIF-pooling layers, with the objective of extracting and shrinking the representation of the features. It is worth mentioning that we have altered the pooling strategy used in the existing CSNN model so that the input from different pixels within a patch is taken into account when creating spike-based feature maps. More information can be found in Section 4.2.

Due to the growth of the data dimension caused by GAFR coding, it can be difficult for conventional Convolutional Spiking Neural Network (CSNN) [23] to accurately identify the relevant and important features in spike image sequences. To this end, an improved Convolutional Block Attention Mechanism (CBAM) is used by assigning its ability to cope with spikes. The biologically plausible characteristic of REAT-CSNN is further enhanced by the Spike-based Spatial Attention Module (SSAM). This module is essential to locate the critical features in the spatial dimension of $I_{k}$, which correspond to the temporal patterns, such as tendency and periodicity, in a time series. Furthermore, the Spike-based Channel Attention Module (SCAM) is used to identify the critical features in the channel dimension of $I_{k}$. This module acts as a filter to focus on the important spatio-temporal correlations in MTS, which are discussed further in Section 4.3.

Lastly, a fully LIF-based connection layer is stacked, giving the final output of the forecast result in the next single time step T+1.

### 4.1. GAFR Coding

Recently, SNN-based approaches have achieved promising results in computer vision tasks. Since excitatory neurons in SNN models only have the ability to deal with spike information, neural coding schemes are usually used to convert pixels in an image into spikes before feeding the image into SNN models. Considering the maturity of these image-driven SNN-based methods, a barrier to applying these approaches in MTS analysis is how to effectively transfer MTS into images while simultaneously maintaining critical knowledge such as the spatio-temporal correlations between time series. A popular approach is to transform $X_{1:T}$ into an image of size T×N. This image can then be converted into a spike image using temporal or rate coding. However, this is not an accessible method as temporal coding or rate coding has drawbacks, such as high latency, information loss, and data distortion, which limits the ability to represent all of the patterns and dependencies in MTS.

In this paper, inspired by the fact that GAF is a useful technique for transferring time series into images, such that a CNN model can be used for time series analysis, we combined GAF and rate coding to create an integrated coding scheme called GAFR coding. This scheme takes advantage of both GAF and rate coding as GAF has been demonstrated to maintain the temporal information of time series, while rate coding is more effective than temporal coding in preserving the original features of images.

The GAFR coding scheme consists of two steps. First, $X_{1:T}$ is transformed into GAF data with multiple channels, where each time series $X_{n\in N}$ is treated as a channel. In this case, the MTS is converted into a multi-channel image. Second, the obtained multi-channel image data are encoded using rate coding to enable spike-based computation. Each pixel value of the image data is represented as the firing rate of a spiking neuron in the spike-based computation.

### 4.2. LIF-Pooling

In CNN, a pooling layer is used to compress a set of pixels into a compact representation. The pooling operation has the benefit of reducing the feature dimension so that the size of the model parameters can be decreased. Meanwhile, it generalizes the features extracted by convolutional filters with little information lost since the features in a patch are usually correlated.

The situation differs from that above when employing the pooling operation on spike images. To begin with, either the input or output from the medium layers of an CSNN model is an image in which the intensity of each pixel is a binary value. In this case, suppose a max-pooling is used, then the output is always 1 if there exists at least one active spike in a patch. It not only aggravates energy consumption, but also makes inefficient event-driven triggering more likely, further resulting in more time being required for model learning. In addition, the temporal correlation over different time lags is encoded in each pixel in the spike image, leading to the features independently distributed in each pixel. Therefore, it easily filters out critical spikes after the max-pooling operation. Unfortunately, although average-pooling can alleviate the issue caused by max-pooling by leveraging the information from all of the pixels in a patch, a threshold has to be used to control the output to be a binary value, and such a threshold is an empirical value that is hard to precisely set.

Inspired by LIF-based convolution, we developed an event-driven pooling mechanism. As shown in Figure 2, the block in the middle of the figure refers to the proposed LIF-pooling module, which consists of two components. In the first component, each patch is flattened as a one-dimensional tensor. Each tensor is fed to the second component that plays the same function as an LIF neuron that has an output estimated based on Formula (Equation 6).

Given an intermediate spike-based feature image I, the number of spikes activated after the LIF-based pooling with the membrane potential threshold $u_{lp}$ in a region of interest $\tilde{I}I$ is denoted by $H(\tilde{I})$, and the number of spikes after average-pooling with the configured threshold $\epsilon_{ap}$ is represented by $\bar{H}\left(\tilde{I}\right)$. This leads to the following theorem:

**Theorem** **1.**
*$H\left(\tilde{I}\right)\geq \bar{H}\left(\tilde{I}\right)$ when $\tau_{m}$ is equal to the size of a patch and $u_{lp}=\epsilon_{ap}$.*


**Proof.** We divided $\tilde{I}$ into a set of patches *P*, where each patch p∈P has equal size. We estimated $H(\tilde{I})$ as follows:(12)$$H\left(\tilde{I}\right)=\sum_{p\in P}o_{p}\left(t\right),$$
where $o_{p}\left(t\right)$ is estimated based on $u_{p}\left(t\right)$ with Formulas (Equation 5) and (Equation 6). In average-pooling, given a patch *p*, we first calculated the sum of intensities in *p* with(13)$${\tilde{u}}_{p}=\frac{1}{S}\sum_{s\in S}p_{s},$$
where *S* is the size of *p*. Suppose $\tau_{m}=S$, then we substitute (Equation 13) into (Equation 5) and $u_{p}\left(t\right)$ is rewritten as(14)$$u_{p}\left(t\right)=\left(1-\frac{1}{S}\right)u_{p}\left(t-1\right)+\frac{1}{S}u_{rest}+{\tilde{u}}_{p}.$$Taking into account that S>1, $u_{rest}>0$, and $u_{p}\left(t-1\right)>0$, we have $u_{p}\left(t\right)>{\tilde{u}}_{p}$. Given $u_{th}=\epsilon_{ap}$, it is reasonable to believe that many more spikes will be activated after LIF-based pooling compared to average-pooling. We have the following three cases:
Case 1: $u_{p}\left(t\right)\geq \epsilon_{ap}\geq {\tilde{u}}_{p}$,Case 2: $\epsilon_{ap}\geq u_{p}\left(t\right)>{\tilde{u}}_{p}$,Case 3: $u_{p}\left(t\right)>{\tilde{u}}_{p}\geq \epsilon_{ap}$.Accordingly, there should be at least one patch p∈P in which one of the above three cases is satisfied. Therefore, $H\left(\tilde{I}\right)\geq \bar{H}\left(\tilde{I}\right)$ is proven. □

The theorem suggests that the risk of data loss when using average-pooling in a region of interest of a spike feature map can be reduced via using LIF-pooling under certain conditions. Additionally, it provides us with a guide to configure the thresholds ($\tau_{m}$ and $u_{lp}$) used in LIF-pooling. The effectiveness of how well the information is preserved after applying our proposed pooling mechanism is visualized in Section 5.4.1.

### 4.3. SCBAM

The proposed Spike-based Convolutional Block Attention Mechanism (SCBAM) is a variant of CBAM developed by Woo et al. [51]. Similarly to CBAM, SCBAM also consists of two modules, namely SCAM and SSAM, and it is implemented in spatial and channel dimensions, respectively. The major difference between SCBAM and CBAM is that SCBAM enables CBAM to operate on the intermediate spike-based feature map with the output in the format of a spike rather than real numbers. Therefore, the information propagated through all layers of the REAT-CSNN model is spike-based. Furthermore, the experimental results show that SCBAM performs similarly to CBAM. To better illustrate, we take SCAM as an example, which is shown in Figure 3. More precisely, to enhance the event-driven capability of SCBAM, we implement SCAM and SSAM in a sequence of discrete time windows. In each time window Δ, SCAM and SSAM are performed according to the definition in Formulas (Equation 8) and (Equation 9):(15)$$\begin{array}{c} SCAM\left(I_{\Delta }\right)≜LIP\left(LP\left(I\right)\right), \end{array}$$(16)$$\begin{array}{c} SSAM\left(I_{\Delta }^{\prime }\right)≜LIF-Conv\left(LP\left(I^{\prime }\right);LP\left(I^{\prime }\right)\right), \end{array}$$
where MLP, AvgP, and MaxP in (Equation 8) are replaced by LIF and the LIF-pooling (LP) developed in Section 4.2, and the Conv in (Equation 9) is replaced by LIF−Conv.

Unfortunately, the above modification is not powerful enough to guarantee that SCBAM is completely implemented on the spikes because of the existence of weighted sum operations. To this end, we perform the AND (∪) and OR (∩) operations instead of the weighted sum operations as follows:(17)$$\begin{array}{c} I_{\Delta ,1}^{\prime }=SCAM\left(I\right)\cup I, \end{array}$$(18)$$\begin{array}{c} I_{\Delta ,2}^{\prime }=SCAM\left(I\right)\cap I, \end{array}$$(19)$$\begin{array}{c} {\tilde{I}}_{\Delta ,1}=SSAM\left(I_{\Delta ,1}^{\prime }\right)\cup I_{\Delta ,1}^{\prime }, \end{array}$$(20)$$\begin{array}{c} {\tilde{I}}_{\Delta ,2}=SSAM\left(I_{\Delta ,2}^{\prime }\right)\cap I_{\Delta ,2}^{\prime }. \end{array}$$Hereafter, we concatenate ${\tilde{I}}_{\Delta ,1}$ and ${\tilde{I}}_{\Delta ,2}$ with(21)$${\tilde{I}}_{\Delta }={\tilde{I}}_{\Delta ,1}∥{\tilde{I}}_{\Delta ,2},$$
and reduce the dimension of ${\tilde{I}}_{\Delta }$ using one-dimensional convolution.

## 5. Experimental Results and Analysis

### 5.1. Tasks and Data Sets

We evaluate the performance of the proposed REAT-CSNN model by applying it in a variety of MTS prediction tasks, which are briefly introduced as follows. All datasets are available via https://github.com/Yongsheng124/REAT-CSNN (accessed on 13 March 2025).

**Stock prediction**: We predict the direction of stock prices by taking into account the correlation between different stocks. For example, the stocks in the same sector, such as banking or healthcare, are likely to move in the same direction and react to the market in a similar way. To this end, we collected data from five gaming companies (Activision, Blizzard, Electronic Arts, Nintendo, and Tencent) over a period of 3094 days, from 4 January 2010 to 4 April 2022. The data set includes the daily date, opening price, closing price, the highest price, the lowest price, and the transaction volume, totaling 15,470 records. To standardize the prediction across all five companies, we normalized each feature in the data set. Specifically, our prediction objective was to use the all of the features from the first nine time steps to forecast the closing price at the tenth time step.**PM2.5 prediction**: We predict the rise and fall of PM2.5 levels in Beijing using the PM2.5 data [52] provided by the US Embassy in Beijing from 1 January 2010 to 31 December 2014, which consists of 43,824 records. This data set includes 12 features such as the year, month, day, hour, dew point, temperature, PM2.5 concentration, wind direction, air pressure, wind speed, snowfall, and precipitation. For convenience, we integrate year, month, day, and hour into a time feature, and, for missing values, we delete the entire record. Similarly, we use data from every nine time steps to predict the PM2.5 concentration for the next time step.**Air quality prediction**: We predicted the rise and fall of concentration $NO_{x}$ to monitor air quality using the data set collected by the air quality chemical multisensor device deployed on highly polluted roads in an Italian city from March 2004 to February 2005 [53]. This data set contains information on the average hourly concentrations of five types of polluting gases, including CO, hydrocarbons, benzene, and nitrogen oxides. The data set has 9358 observations, including 15 features such as time, temperature, relative humidity, absolute humidity, and polluting gas concentration. Moreover, we directly delete records with missing values.**Air pollution prediction**: We predict the rise and fall of concentration $NO_{x}$ to track air pollution using the gas turbine CO and the emission data set. This data set contains 36,722 observations collected by 11 sensors located in northwest Turkey. The data span a period of 5 years, from 1 January 2011 to 31 December 2015, with a sampling frequency of one hour. Each record includes ambient temperature, air pressure, humidity, air filter difference pressure, gas turbine exhaust pressure, turbine inlet temperature, turbine after temperature, compressor discharge pressure, turbine energy yield, CO concentration, and $NO_{x}$ concentration, totaling 11 features. It is worth noting that each observation does not provide the time feature, and it still contains the inherent time feature because it is strictly sorted by time. As above, we delete those records with missing values.

### 5.2. Experimental Setting

The data set is split into three sections: 80% for training, 10% for validation, and 10% for testing. All data records are normalized to a range of 0 to 1. Data cleaning is used to process missing or noisy data points, and we use features from nine time steps to predict the rise and fall of the target feature for the next time step. In addition, to minimize the impact of randomness, we conducted each experiment 10 times with different random seeds, assuming the results followed a standard normal distribution, and we then calculated the mean μ and standard deviation σ to report the model’s final result as μ±σ.

We used RNN-based and GNN-based approaches, which have been widely used for MTS prediction, as baselines. Furthermore, we compared our method with another SNN model, namely Long-Short-Term Memory SNN (LSNN). The details of these baselines are as follows.

RNN: A neural network with memory ability, allowing the network to capture temporal correlations in the sequence. We used a simple one-layer RNN model.LSTM: A variant of RNN that introduces a gating mechanism with the advantage of handling long-term dependencies. In this paper, we used a simple one-layer LSTM model.GRU: Another variant of RNN, which has a higher computational efficiency than LSTM. Similar to RNN and LSTM, we used a one-layer GRU model.GCN [54]: Operates directly on graph-structure-based data that are built on the basis of the relationships between stocks.LSNN [55]: Integrates a neuronal adaptation mechanism into the recurrent SNN (RSNN) model with the function of capturing dynamic processes on large time scales, including the excitability and inhibition of neurons, spike frequency, spike time interval, etc.

We used Adam as the optimizer for REAT-CSNN, with a mean squared error (MSE) loss function and a batch size of 20. The learning rate was set to $1\times 10^{-4}$. To control variables, we used the same hyperparameters for all other benchmark models and conducted 10 experiments in the same environment. We evaluated the performance of the model by measuring the single-step prediction accuracy that is calculated by(22)$$Accuracy=\frac{\sum_{n\in N}\phi \left({\hat{X}}_{n,T+1},X_{n,T+1}\right)}{N},$$
where ${\hat{X}}_{n,T+1}$ is the result predicted for time series *n* at the time step T+1, and $\phi ({\hat{X}}_{n,T+1},X_{n,T+1})=1$ if ${\hat{X}}_{n,T+1}=X_{n,T+1}$, or 0 otherwise. All models were run on Pytorch 1.10.2, and the hardware used was an Intel(R) Platinum 8255C CPU (2.50 GHz), 40G memory, and an RTX2080Ti (the code is available via https://github.com/1422819414/REAT-CSNN (accessed on 13 March 2025)).

### 5.3. Results

The results in Table 1 demonstrate that REAT-CSNN performed better on all tasks, apart from the prediction of air pollution. Our model surpasses LSNN with an improvement of up to 3.2% in PM2.5 prediction. Additionally, our model performs better than typical ANN- and graph-based MTS predictive methods. Compared to LSTM and GRU, the better performance achieved by our proposed model in different tasks indicates that the event-driven architecture of REAT-CSNN has the capacity to capture long- and short-term patterns in MTS. In particular, under a comparable parameter scale, the performance gap between our model and RNN can reach almost 3% in stock prediction. In PM2.5 and air quality prediction, our model outperforms RNN with an improvement of 0.3% and 0.9%, respectively. Furthermore, the smaller variance suggests that our model has a more stable performance than RNN.

It is noteworthy that REAT-CSNN outperformed GCN in stock prediction. As previously mentioned, GCN has a higher predictive accuracy than other ANN-based approaches due to its ability to capture explicit and implicit relationships between stakeholders in a stock, which enhances the predictive accuracy. Notably, our model outperformed GCN, improving by 0.33%, with only 1/5 of the number of parameters. This demonstrates that our proposed model has a similar capability to GCN in capturing deep spatio-temporal correlations between stocks. However, REAT-CSNN does not require any prior knowledge to construct the graph that is used in GMN-based methods.

Figure 4 shows the changes in the test set accuracy during the training process of each model. It can be seen that, when facing the air quality data set with a not strong non-linear complexity between variables, our model has an advantage. However, REAT-CSNN is slightly inferior to LSNN in terms of air pollution prediction. This is mainly due to the fact that the convolution operation and SCABM component do not have as much of an impact as they do in other MTS prediction tasks since the data collected from different detectors have a weak spatio-temporal correlation due to the large physical distance. Therefore, our proposed model is more suitable for MTS where variables interact simply directly and exhibit minor nonlinear relationships. Although convolution and SCBAM enhances the ability to capture complex nonlinear relationships to some extent, it still cannot compensate for the structural differences compared to RNN-based methods. Similarly, as depicted in Figure 4b, the accuracy of GCN in dealing with air pollution prediction keeps oscillating around 0.5 and basically fails to train. This indicates that convolution-based methods find it difficult to extract overly complex, high-dimensional, and spatio-temporal relationships. Furthermore, it is reasonable to believe that REAT-CSNN is comparable to ANN- and GNN-based methods in terms of MTS prediction, particularly when there is a strong spatio-temporal correlation between time series variables but with a lower energy requirement (see Section 5.5).

### 5.4. Ablation

We evaluated the effectiveness of the proposed LIF-pooling and SCABM through ablation tests. More precisely, to test the effectiveness of the LIF-pooling strategy, we first carried out an ablation test on the stock prediction task, the results of which are shown in Figure 5. These results demonstrate the difference between average-pooling (where the threshold is 0.5) and LIF-pooling. Then, we conducted a comparison between the max-pooling and LIF-pooling strategies, as shown in Table 2, to further illustrate the power of LIF-pooling. Furthermore, two sets of ablation tests are given in Table 2, which were performed on PM2.5 and stock data sets, respectively. The tests were designed to assess the effect of different attention modules on the model’s performance by implementing the following configurations:REAT-CSNN without any attention modules;REAT-CSNN when only considering SSAM;REAT-CSNN when only considering SCAM;REAT-CSNN by switching the implementation orders of SSAM and SCAM in SCBAM.


Figure 5Comparison between LIF-pooling (LP) and average-pooling (AP).
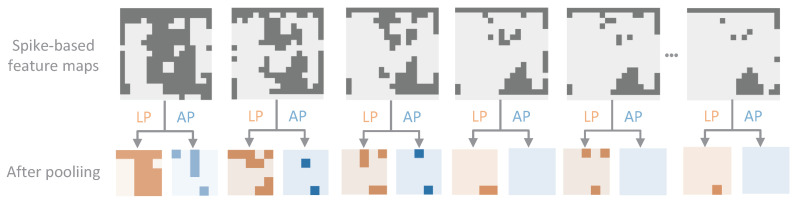

biomimetics-10-00240-t002_Table 2Table 2Ablative analysis on the max-pooling and attention modules.Attention ModuleStockPM2.5SCAMSSAMMax-PoolingLIF-PoolingMax-PoolingLIF-Pooling✗✗


52.86±0.43%




52.78±0.45%




58.71±0.71%




59.74±0.33%

✗✓

52.57±0.19%



53.36±0.41%



59.01±0.79%



59.66±0.38%

✓✗

52.99±0.31%



53.75±0.74%



59.92±1.11%



59.86±0.33%



⇌



52.67±0.41%



53.42±0.57%



58.56±1.29%




60.41±0.39%


✓✓

52.85±0.44%




53.49±0.53%




59.22±1.23%



60.45±0.38%




#### 5.4.1. Pooling Strategy

As shown in Figure 5, we chose six regions from the spike-based feature maps produced by different LIF-Conv layers. We visualize each feature map by representing the activated spikes in deep gray and the non-active spikes in light gray. The orange and blue feature maps are the results of applying LIF-pooling and average-pooling to the selected feature maps, respectively. It is evident that LIF-pooling (orange) preserves much more information than average-pooling (blue). This is in agreement with the analysis provided in Theorem 1. Examining the results of REAT-CSNN in Table 2, which is when max-pooling and LIF-pooling were employed, it is clear that LIF-pooling was highly successful in enhancing the prediction accuracy in the two data sets. This verifies that the proposed LIF-pooling mechanism is much more effective than the max-pooling operation.

In addition to the observation mentioned above, our experimental results with max-pooling reveal some interesting findings. Surprisingly, the prediction accuracy of our model with an attention mechanism does not improve when max-pooling is used. In fact, the accuracy of the model even decreased in some cases. For example, the average accuracy of stock prediction with max-pooling and without attention modules is 52.86%, which is slightly higher than the 52.85% obtained when using the SCAM and SSAM modules. Similar results were observed in the PM2.5 prediction. As discussed in Section 4.2, having too much information does not necessarily mean that more precise features are extracted. Therefore, although max-pooling assigns a value of 1 to any element in a patch of the spike feature map, which results in too much information being retained, it makes it difficult for the attention mechanism to assign the correct weights to the truly important features. The results obtained from REAT-CSNN with the application of LIF-pooling are distinct from those of the above (in the second and last columns). The attention mechanism is capable of increasing the accuracy of the prediction.

#### 5.4.2. Attention Modules

From an application point of view, SCAM and SSAM play different roles in predicting MTS. Notably, SCAM had the most significant improvement in stock prediction, suggesting that channel attention is more critical than spatial attention for accurate stock prediction as it can detect and pinpoint the essential spatio-temporal correlation between time series variables. On the other hand, both modules have a clear improvement in the accuracy of PM2.5 prediction. There is only a minor variation (of 0.07% and 0.04%) when the two modules were exchanged in the REAT-CSNN model. These results demonstrate that the proposed attention modules are successful in improving the forecasting performance in MTS tasks.

### 5.5. Energy Efficiency

Low energy consumption is the most notable characteristic of SNN-based models. To this end, we estimate the energy consumption of our proposed model with the aforementioned baselines. In our model, most of the energy consumption occurs in LIF-based convolution layers and LIF-based FC layers. Unlike traditional CNNs, where the computational complexity of a convolution heavily depends on the number of Multiplication and Addition Computing (MAC) operations [56], LIF-Cov is composed of several Addition Computing (AC) operations due to its event-driven activation nature. Therefore, we estimate the energy consumption at each LIF-Cov layer ($E_{LC}$) as follows:(23)$$E_{LC}=C_{in}\times k^{2}\times C_{out}\times W\times H\times E_{AC},$$
where $C_{in}$ is the channel size of the input; *k* is the kernel size; $C_{out}$ is the channel size of the output; *W* and *H* represent the width and height of the feature map, respectively; and $E_{AC}$ is the energy consumption of a unit AC operation. More specifically, $E_{AC}=0.9$ pJ when AC is operated on 32-bit floating numbers, and $E_{AC}=0.1$ pJ when AC is operated on 32-bit integers [56]. The energy consumption at each LIF-FC layer ($E_{LF}$) is estimated as follows:(24)$$E_{LF}=N_{in}\times N_{out}\times E_{AC},$$
where $N_{in}$ and $N_{out}$ are the number of input and output neurons, respectively. Based on $E_{LC}$ and $E_{LF}$, we can further estimate the energy consumption of REAT-CSNN and LSNN, which are denoted by $E_{REAT-CSNN}$ and $E_{LSNN}$.

The energy consumption of RNN, LSTM, and GRU is denoted by $E_{RNN}$, $E_{LSTM}$, and $E_{GRU}$, which is based mainly on the energy consumption in each recurrent layer, denoted by $E_{RL}$ and estimated as(25)$$E_{RL}=\left(D_{in}+D_{h}\right)\times D_{h}\times T\times S_{nl}\times E_{MAC}.$$In (Equation 25), $D_{in}$ and $D_{h}$ are the dimensions of the input layer and the hidden layer; $S_{nl}$ is the number of non-linear blocks with $S_{nl}=1$ for RNN, $S_{nl}=4$ for LSTM, and $S_{nl}=3$ for GRU; and $E_{MAC}$ is the energy consumption of a unit MAC operation, which is equal to 4.6 pJ when MAC is operated on 32-bit floating numbers and 3.2 pJ when MAC is operated on 32-bit integers.

The energy consumption of GCN ($E_{GCN}$) is estimated as(26)$$E_{GCN}=2\times N_{V}\times N_{E}\times \left(2\times D_{V}\right),$$
where $N_{V}$ is the number of vertices, $N_{E}$ is the number of edges, and $D_{V}$ is the feature dimension at each vertex.

Table 3 presents the energy consumption of different models from the perspectives of floating point operations per second (FLOP), operations on 32-bit floating numbers (Float32), and operations on 32-bit integers (Int32). It is evident that our model has significantly lower energy consumption than traditional deep learning models, particularly for GRU and LSTM. As illustrated in Figure 6, the decrease in energy consumption is up to two orders of magnitude less than the LSTM, GRU, and GCN operated on FLOP and Int32. However, the lowest energy consumption is from the LSNN. This is because REAT-CSNN has more components, such as the attention mechanism, which are designed to improve prediction accuracy.

## 6. Conclusions

We developed a REAT-CSNN model that utilizes a GAFR coding scheme, a LIF-pooling strategy, and a SCBAM to improve the accuracy of MTS prediction. This model is more biologically plausible as it enables all components to be implemented on the basis of spikes instead of real numbers. Additionally, it has a strong capability in exploring and capturing the spatio-temporal correlations in MTS. In the future, we will attempt to expand our work to real-world scenarios by taking into account the dynamics of spatio-temporal correlations. In addition, the model will be tested on SNN-enabled hardware.

## Figures and Tables

**Figure 1 biomimetics-10-00240-f001:**
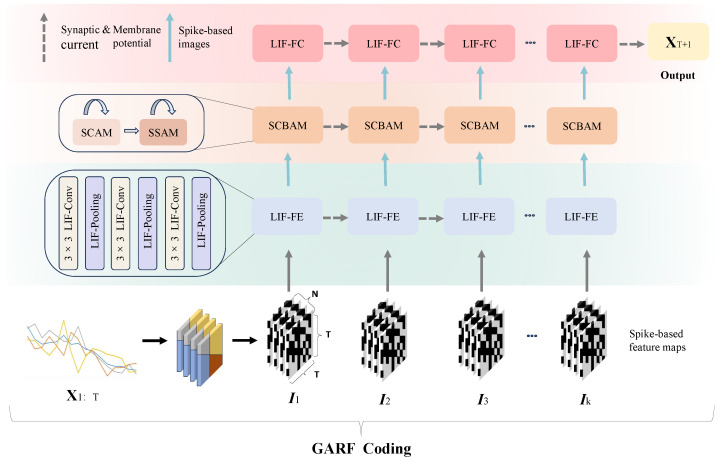
The architecture of REAT-CSNN.

**Figure 2 biomimetics-10-00240-f002:**
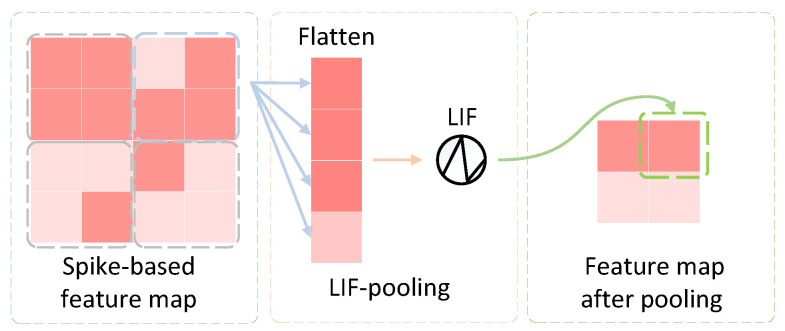
The principle of LIF-pooling.

**Figure 3 biomimetics-10-00240-f003:**
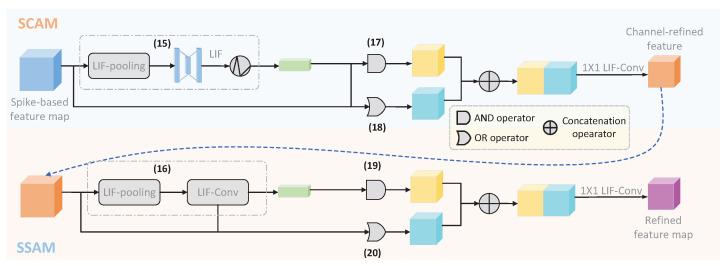
The process of SCBAM in one time step.

**Figure 4 biomimetics-10-00240-f004:**
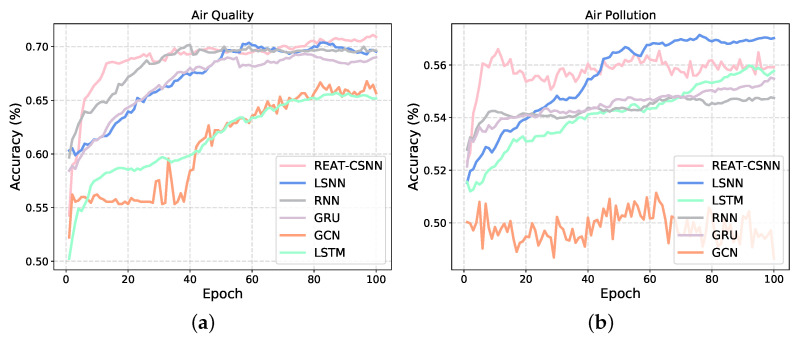
The training process of REAR-CSNN and other models on different data sets. (**a**) On the air quality data set. (**b**) On the air pollution data set.

**Figure 6 biomimetics-10-00240-f006:**
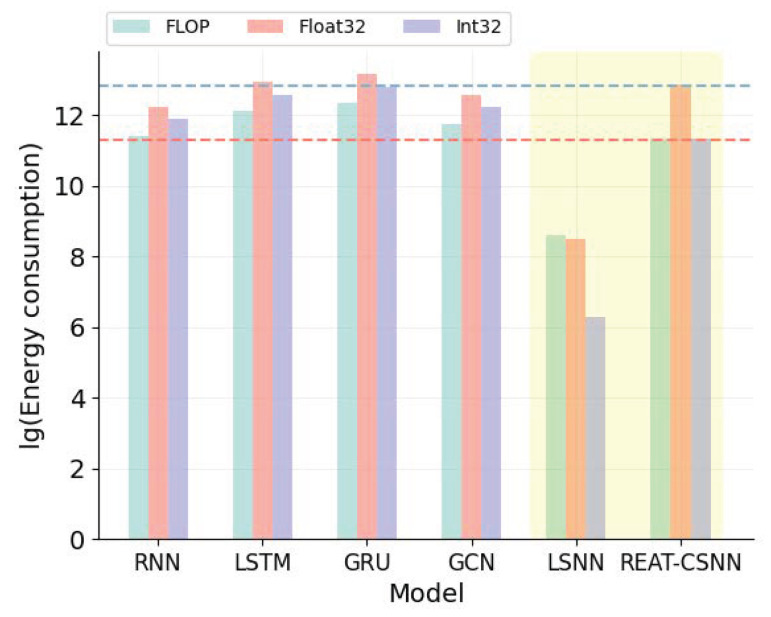
Energy consumption analysis.

**Table 1 biomimetics-10-00240-t001:** The accuracy of MTS prediction on multiple data sets.

Network	Param (K)	T	Data Sets			
			Stock	PM2.5	Air Quality	Air Pollution
RNN (one-layer)	3.54	9	50.26±0.22%	59.86±0.33%	69.84±0.89%	54.76±0.71%
LSTM (one-layer)	5.95	9	53.13±0.48%	58.89±0.28%	65.72±0.55%	55.21±0.68%
GRU (one-layer)	4.48	9	53.07±0.28%	59.57±0.37%	68.80±0.59%	54.96±0.32%
GCN [54]	15.92	9	53.16±0.78%	51.10±0.19%	65.93±0.88%	51.15±0.44%
LSNN [55]	3.6	9	52.20±0.42%	56.91±0.16%	69.63±0.43%	56.72±0.23%
REAT-CSNN (ours)	3.57	9	53.49±0.53%	60.12±0.17%	70.79±0.16%	55.45±0.43%

**Table 3 biomimetics-10-00240-t003:** Energy consumption analysis.

Model	FLOP	Float32	Int32
$E_{RNN}$	89,984	206,963.2	143,974.4
$E_{GRU}$	179,968	413,926.4	287,948.8
$E_{LSTM}$	224,960	517,408	359,936
$E_{GCN}$	126,624	291,235.2	201,667.2
$E_{LSNN}$	**5377**	**4839.3**	**537.7**
$E_{REAT-CSNN}$	84,714.3	381,214.35	84,714.3

## Data Availability

The data are contained within the article.

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
