# Peer review of "A Reinforced, Event-Driven, and Attention-Based Convolution Spiking Neural Network for Multivariate Time Series Prediction"

_biomimetics, 2025, doi:10.3390/biomimetics10040240_

Round 1

Reviewer 1 Report

Comments and Suggestions for Authors

- A major concern of this manuscript is the computational experiments. The authors need to provide a clear and complete detail on computational experiments. 

- Because there are several datasets used for the validation, the authors need to provide further detail corresponding to each dataset in addition to a brief description given in Section 5.1. 

- The authors should provide some figures (such as time series or waveform) depicting successful and failure cases in addition to the numeric metrics. 

- The list of keywords does not meaningfully represent the manuscript. This should be revised. 

- The abstract can also be improved. 

- The manuscript still lacks the comparative study. 

Author Response

Thank you very much for your valuable suggestions. Our reply is attached. Please review it. Thank you again!

Reviewer 2 Report

Comments and Suggestions for Authors

This manuscript introduces a novel convolution spike neural network (SNN) for multivariate time series (MTS) prediction. There are three key innovations of this work. A joint Gramian Angular Field and Rate (GAFR) coding technique to convert MTS data into spike images while preserving temporal and spatial correlations. An improved Leaky Integrate-and-Fire (LIF) based pooling strategy designed to enhance feature retention compared to traditional max and average pooling methods. A redesigned attention mechanism tailored for spike-based input, strengthening event-driven characteristics in weighting operations. The advantages of using those techniques to improve the performance of their SNN model are demonstrated from experiments on datasets related to stock market prediction and environmental monitoring (PM2.5, air quality, and air pollution). Their proposed model demonstrates comparable or superior performance to CNN, RNN, and GNN-based methods while consuming significantly less energy. Thus, I recommend the publication of this work with some minor suggestions.

  1. In section 3 and 4, some notations and equations (for example, in the GAFR coding and SCBAM descriptions) are densely packed, which might make it challenging for a reader unfamiliar with the topic. I was wondering if the authors could add more references to related literature for readers to further understand them.
  2. In Table 2, when comparing the accuracy of MTS prediction, there are confidence intervals for results. I was wondering if the authors could explain more about how they estimate this.
  3. In section 5, REAT-CSNN did not outperform LSNN for air pollution dataset. I was wondering if the authors could discuss more about what kind of dataset is more suitable for their model.
  4. I was wondering if the authors could add the discussion about the computational cost and training time of various models for readers to see how their model is compared to other models.
  5. I was wondering if the authors could discuss whether the better performance of their model could come from the more complex structure and whether increasing the number of layers could improve the performance of RNN, LSTM, and GRU.
  6. Minor text issues:
    1. In abstract line 3, the full name of MTS is not given.
    2. In line 37, again, the full name of MTS is not given.
    3. In line 505, reference is missing for ‘As illustrated in Fig ??’

Author Response

Thank you very much for your valuable suggestions. Our reply is attached. Please review it. Thank you again !

Round 2

Reviewer 1 Report

Comments and Suggestions for Authors

The manuscript was significantly revised. It is now in an acceptable form.